# A Load Balancing Routing Mechanism Based on SDWSN in Smart City

**Xin Cui [1,2], Xiaohong Huang [1], Yan Ma [1,\*] and Qingke Meng [2]**

[1]  Institute of Network Technology, Beijing University of Posts and Telecommunications, Beijing 100876, China; cxsd2007@163.com (X.C.); cxsd@bupt.edu.cn (X.H.)

[2]  College of Computer Science and Technology, Shandong University of Technology, Zibo 255000, China; cx@sdut.edu.cn

\*  Correspondence: mayan@bupt.edu.cn

**Abstract:** In the wireless sensor network infrastructure of smart cities, whether the network traffic is balanced will directly affect the service quality of the network. Because of the traditional WSN (wireless sensor network) architecture, load balancing technology is difficult to meet the requirements of adaptability and high flexibility. This paper proposes a load balancing mechanism based on SDWSN (software defined wireless sensor network). This mechanism utilizes the advantages of a centralized control SDN (software defined network) and flexible traffic scheduling. The OpenFlow protocol is used to monitor the running status and link load information of the network in real time. According to the bandwidth requirement of the data flow, the improved load balanced routing is obtained by an Elman neural network. The simulation results show that the improved SDSNLB (software-defined sensor network load balancing) routing algorithm has better performance than LEACH (Low Energy Adaptive Clustering Hierarchy) protocol in balancing node traffic and improving throughput.

**Keywords:** WSN; SDN; Elman neural network; load balancing

---

## 1. Introduction

With the rapid development of smart cities, the requirements for the resource allocation mechanism of wireless sensor networks are gradually increasing. In [1], wireless sensor networks are the infrastructure of many fields in smart cities, including automotive electronics, avionics, building automation and industrial automation. In [2], Over the past two decades, various sensing system environments have been designed and deployed in cities, towards the realization of so-called smart cities. Such systems are based on dedicated sensor nodes, as well as ubiquitous but not dedicated devices such as smart phones and vehicle sensors. Due to the dynamic changes of the sensor network structure, network resources (including the remaining energy of the nodes, available bandwidth, etc.) are also constantly changing, and the requirements for the adaptability of routing protocols are proposed. In particular, in wireless sensor networks, the routing protocol is not only the resource allocation of a single node, but also the resource balancing problem of the overall network. In [3], the architecture of a smart city can be divided into four layers, including sensing, networking, cloud computing and applications. In the future, smart cities and smart sensors will be built based on sensor technology, hybrid networks, cloud computing and storage, and big data analysis. By achieving the load balancing of resources, the QoS (quality of service) can be improved and the lifetime of the whole network can be prolonged. Therefore, it is an urgent problem to improve the routing optimization of the communication link, enhance the efficiency of network deployment, and realize network load balance in smart city sensor networks.

The problem of load balancing in WSN has been studied by a large number of scholars for a long time. In order to solve the problem of load imbalance in wireless sensor networks, researchers put forward many solutions. In [4], Based on LEACH, an improved LEACH protocol in WSNs is proposed. The LEACH protocol acts as a low-power adaptive hierarchical routing protocol, randomly selecting cluster heads in a cyclical manner to evenly distribute energy consumption in the network. In WSN, sensor nodes depend on each other when forwarding information packets from a source station to a base station by routing algorithm. Some routes may be better than others, which may result in an unbalanced contention for different routes, as one route may be exhausted more often or faster than others. In [5], they analyze this problem from the perspective of game theory and model the path selection problem in WSN as an evolutionary anti-coordination routing game. In order to prolong network lifetime, candidate forwarders should be selected so that load is balanced among nodes. In [6], they proposed an ORR, an opportunistic routing protocol. In [7], they proposed a simple and effective queue utilization based RPL (QU-RPL) that achieves load balancing and significantly improves the end-to-end packet delivery performance compared to the standard RPL. QU-RPL is designed for each node to select its parent node considering the queue utilization of its neighbor nodes as well as their hop distances to an LLN border router (LBR), owing to its load balancing capability. In [8], a framework is proposed for load balanced routing in WSN. In-network path tagging is used to monitor the network traffic load of nodes. Based on this, nodes are identified as being relatively overloaded, balanced or under loaded. A mitigation algorithm finds suitable new parents for switching from overloaded nodes. The routing engine of the child of the overloaded node is then instructed to switch its parent.

The above dynamic routing data forwarding mainly relies on the local node information, which easily falls into the local optimum, while the global improved routing algorithm is difficult to distribute and is not suitable for large-scale sensor networks.

SDN came into being, control and forwarding separation features provide effective way for WSN routing algorithms. As shown in Figure 1, SDN−based network architecture will be more flexible, and application upgrades will be more simplified.

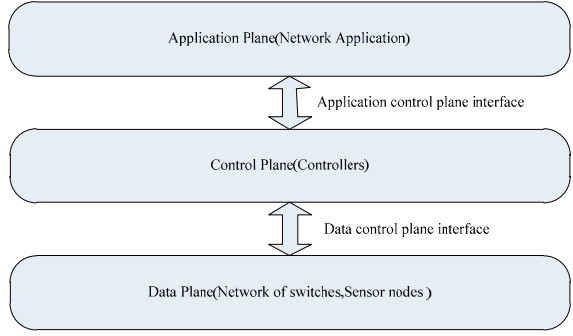

**Figure 1.** Software defined network (SDN) overall architecture.

This paper discusses the WSN architecture based on SDN. For the multi-path optimized SDWSN load balancing problem, a routing scheme based on an Elman neural network for improved solution is proposed. In order to minimize the maximum link utilization in the network, this paper establishes a mathematical model based on multi-path routing technology while satisfying the quality of service, and the model is simulated and verified. The results show that the load balancing strategy can minimize the maximum link utilization in the network, and fully utilize the network link resources to reduce intra-domain delay and extend network lifetime. This paper summarizes and analyzes the SDN technology, and designs the load balancing model of the data plane and control plane in the SDWSN, which provides a new solution for the load balancing problem. Our major contribution can be summarized as follows:

The algorithm model combines the hierarchical structure of an Elman neural network with the clustering structure of a wireless sensor network routing protocol, and designs an Elman neural network model in each cluster structure. By designing a neural network algorithm model, a little amount of data information reflecting the characteristics of the original data is obtained. The fused feature data is transmitted to the controller node for quickly performing load balancing of the sensor node link.

We analyze and derive the proposed traffic matrix clustering algorithm, which describes the dynamic adaptive adjustment process. The central controller performs a cluster analysis and adjustment according to the flow matrix of each controller, quickly combining the sensor node traffic data to determine the output of the improved path matrix.

In addition, because the fairness of the path selection strategy is very important to load balancing, we use the principle of max-min to measure the uniformity of traffic. The max-min means that when there are multiple routing schemes, the routing scheme with the smallest bandwidth utilization in the network is selected.

The rest of the paper is organized as follows: related work about routing protocol to resolve traffic load is summarized in Section 2. In Section 3, system framework and problem analysis are described. Section 4 formulates the traffic load allocation with end to end multipath, proposes the SDSNLB algorithm, and the improved solution. Section 5 shows the performance evaluation and simulation. In Section 6, a summary conclusion is derived.

## 2. Related Work

In [9], it is proposed that traditional sensor networks are inflexible due to over-reliance on proprietary services, so data and control separation modes are suitable for wireless sensor networks. They also point out the challenges of this wireless sensor network combined with software-defined networks, as well as the necessary basic design load balancing requirements. In [10], a new mobile network architecture based on OpenFlow to achieve centralized control of energy-aware on demand for load balancing is proposed. In [11], the research results of the software-defined wireless sensor network into actual hardware systems are translated. The OpenFlow switch built a software-defined wireless sensor network platform and runs a traffic-aware routing algorithm on it. The Open DayLight cluster can accurately analyze the data. The flow direction in the network proves that the traffic-aware routing algorithm can improve network utilization and reduce controller response time. Although these new WSN architectures can solve some of the problems of traditional WSN, we should also see that the resource allocation design of software defined wireless sensor networks is not perfect and the research on load balancing needs to be further improved.

In [12], they propose a QoS-based multi-path routing protocol in SDN. In order to meet the QoS requirements of network applications, this protocol designs a multi-path routing protocol based on SDN, which significantly improves the utilization rate of idle resources in the search space. In [13], they present a unified multi-layer architecture with multiple controllers and a dynamic orchestra plane for software defined multi-domain optical networks. In [14], a framework for multi-layer; multi-vendor optical network management using open standards-based SDN is proposed. In [15], a routing algorithm based on SDN technology is proposed, which is used for a routing calculation between AS (Autonomous Systems). The above documents, in the SDWSN network architecture, can solve some unsolved problems of traditional routing protocol, but cannot respond in real time and be dynamic in the aspect of load balancing of traffic.

In [16], a new control system based on the integration of SDN and IoT in smart city environments is proposed. This control system actuates when an emergency happens, modifies dynamically the routes of normal, emergency urban traffic in order to reduce the time that the emergency resources need to get to the emergency area. In [17], a novel routing protocol applied SDN in wireless multi-hop network is proposed. The routing protocol provide a shortest path, disjoint multipath routing for nodes, and its network lifetime is longer. In [18], they highlight the application challenges faced by

wireless sensor networks for surveillance environments, those faced by the proposed approaches, as well as opportunities that can be realized on applications of SDWSN. In the SDWSN network architecture, the above literature focuses on the special functions that should not be ignored when trying to improve the network function, however, the load balancing design of the traffic is insufficient.

In summary, the research results of the routing protocols in the classic wireless sensor networks have been quite abundant. The software-defined routing protocols for new network architectures have also achieved preliminary research results. However, the software defined wireless sensor network technology is still immature and needs to be explored and studied in many aspects. It is of great significance to conduct in-depth research on its routing protocols.

## 3. System Framework and Problem Analysis

We propose a wireless sensor network model based on SDN, as shown in Figure 2. The core idea of this new architecture is to separate the logical control and network forwarding devices in the WSN. The control organization is responsible for abstracting the underlying network device and application logic. It also supports extended deployment through reserved interfaces to achieve flexible control of the entire network. The model adopts a hierarchical network structure, which mainly consists of a data plane, a control plane, and an application plane. The entire network can be divided into multiple sub-areas. Each sub-area may be a particular network formed by the same or different types of terminal sensor nodes. The sensing node is responsible for sensing the collection and forwarding of object information. One or more controllers aggregate, manage, and reconfigure information across multiple network topologies and node behaviors. The data center is mainly responsible for storing and managing the upload information data of each cluster, and providing the terminal application or user with an interface such as a query. The application provides monitoring and other services for different users such as mobile terminals, desktops, and the cloud to provide convenient feedback for each area's information and status.

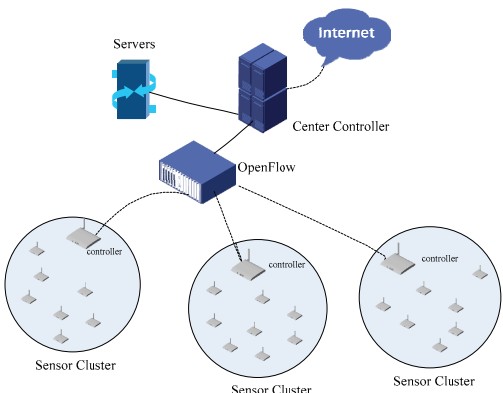

**Figure 2.** Software defined wireless sensor network architecture.

There are three basic roles in this model, namely the master node, the central node, and the normal node. The master node acts as a controller for the entire network structure. This is the central execution unit of programmable control, which controls the real-time status of the network, such as network topology, routing, and routing restrictions, and implements the management and control functions of the entire network. The center node is similar to the OpenFlow switch in SDN, the network device maintains a Flow Table and processes the data stream only through the Flow Table. The generation, maintenance, and delivery of the Flow Table are completely implemented by an external controller. As shown in Figure 3, it is responsible for the matching and forwarding of data streams in wireless sensor networks. A normal user node is only responsible for receiving and executing data streams.

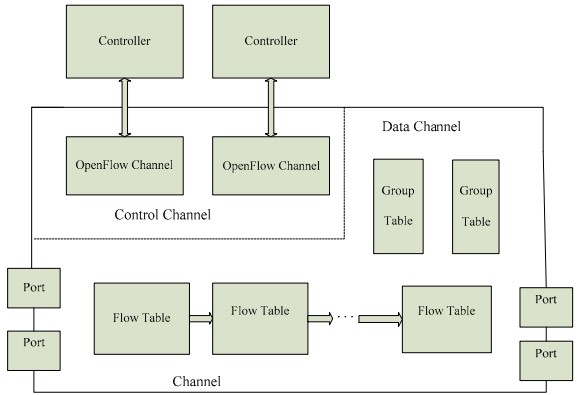

**Figure 3.** Key components of OpenFlow switches.

A major feature of SDN networks is the separation of the control layer and the data layer. The controller can obtain the topology of the entire network. At the same time, it can obtain real-time network status information from the switch, including link load, delay, etc. We can develop a load-balancing routing strategy for forwarding packets at the control layer.

## 4. SDSNLB Algorithm and Solution

In SDWSN, data traffic can be assigned to multiple paths to improve network utility. In this part, we propose end-to-end multipath bandwidth allocation as an optimization problem to achieve efficient use of network resources. In a smart city, load balancing is realized by a multipath transmission of information. Therefore, constraining this problem includes two aspects: multiple paths from source to destination and link bandwidth that makes up each path. The parameters used in our model are summarized in Table 1.

**Table 1.** List of notation.

| Parameters | Meaning |
|---|---|
| V, E | The set of all devices and the set of links between them |
| $C_{(e)}$ | The capacity of link e, e∈E |
| $l_{(e)}$ | The load of link e, e∈E |
| $f_e^{(s,t)}$ | The traffic of between the nodes s,t by link e |
| $TM$ | The traffic matrix |
| $TM_{(s,t)}$ | The traffic from the routing node s to node t |
| $U_{(s,t)}$ | The utilization of link e |
| A | The set of all link utilization |
| m | The amount of utilization contained in A |
| $x_{i,j}$ | The connection weight between the i-th hidden layer (or input layer) to the j-th output layer (or hidden layer) |
| $\beta_j$ | The output value of the jth input layer (or hidden layer) |
| $\hat{y}_i$ | The output value of the i-th output layer |
| $y_i$ | The expected value of the i-th output layer |
| K | The number of expected cluster centers |
| $\theta_N$ | The minimum number of samples in each cluster domain |
| $\theta_S$ | The standard deviation of the sample distance distribution in a cluster domain |
| $\theta_C$ | The minimum distance between two cluster centers. If it is less than this number, the two clusters are merged |
| L | The maximum number of cluster centers that can be merged in one iteration |
| I | The number of iterations |
| $S_j$ | The cluster domain of traffic matrix |
| $N_j$ | The number of traffic matrices in $S_j$ |
| n | Dimension of sample feature vector |

**Definition 1.** *End-to-end path: A transmission line consisting of a series of store-and-forward nodes, transfer data information from the source to the destination.*

**Definition 2.** *Available link bandwidth: Assume that an end-to-end path contains n links, $C_{(e)}$ is the initial capacity of link e. The carried traffic is $l_{(e)}$, Define the available link bandwidth is $C_{(e)} - l_{(e)}$. The available link bandwidth reflects the link characteristics actually transmitted.*

**Definition 3.** *Link Bandwidth Utilization: For $\forall (s,t) \in E$, the bandwidth utilization of the link is $U_{(s,t)} = \frac{l_{(s,t)}}{C_{(s,t)}} \times 100\%$.*

**Definition 4.** *Average utilization of links: The average utilization of links in the network is the average of the bandwidth utilization of all links, defined as $\overline{U_{(s,t)}} = \frac{\sum_{U_{(s,t)} \in A} U_{(s,t)}}{k}$.*

Because the fairness of the path selection strategy is quite important to load balancing, we derive a load balancing objective function by using the fairness principle of Min-Max. Therefore, the objective function of link load balancing is as follows:

Objective function: minimize $y = \frac{\sum_{U_{(s,t)} \in A} \left( U_{(s,t)} - \overline{U_{(s,t)}} \right)^2}{m}$

Subject to:

$$\sum_{i<j(i,j)\in E} \sum f_{(i,j)}^{(s,t)} - \sum_{k>j(j,k)\in E} f_{(j,k)}^{(s,t)} = \begin{cases} -TM_{(s,t)} j = s \\ TM_{(s,t)} j = t \\ 0 j \neq s, t \end{cases}$$

$$l_{(e)} = \sum_{(s,t)\in v*v} f_e^{(s,t)} \leq C_{(e)} e \in E$$

$$f_e^{(s,t)} \geq 0 e \in E; s, t \in v$$

The Elman neural network, which is a typical dynamic recurrent neural network, which adds a receiving layer to the hidden layer as a one-step delay operator to achieve the purpose of memory, thus enabling it. The system has the ability to adapt to time-varying characteristics and enhances the global stability of the network. It has more computing power than the feed forward neural network, and is very suitable for establishing prediction models for nonlinear data such as network traffic time series. However, the traditional Elman network uses the gradient descent method to train the network parameters, but the gradient descent method has the disadvantages of slow convergence and easily falls into local minimum values. Therefore, we use the cluster and parameter adjustment methods to train network parameters, and achieve the goal of rapid optimization.

During Elman's learning of the sample, the error is distributed among the neurons in each layer. According to the error, the least squares method is used to solve the error.

Then there exists $\sum_{j=1}^{n} x_{ij}\beta_j = y_i (i = 1, 2, 3, \ldots, m)$

$$\mathbf{X} = \begin{pmatrix} x_{11} & \cdots & x_{1n} \\ \vdots & \ddots & \vdots \\ x_{m1} & \cdots & x_{mm} \end{pmatrix} \quad \beta = \begin{pmatrix} \beta_1 \\ \beta_2 \\ \vdots \\ \beta_3 \end{pmatrix} \quad \mathbf{y} = \begin{pmatrix} y_1 \\ y_2 \\ \vdots \\ y_n \end{pmatrix}$$

Obviously, Equation $X\beta = y$ is generally unsolvable, but $X\beta = \hat{y}$ is known, so the parameters should be modified from the relationship between the predicted and expected values of the Elman neural network.

$$y_i = \hat{\beta}_0 + \hat{\beta}_1 \times \beta_1 + \hat{\beta}_2 \times \beta_2 + \cdots + \hat{\beta}_n \times \beta_n + e_i$$

$$e_i = y_i - \hat{\beta}_0 - \hat{\beta}_1 \times \beta_1 - \hat{\beta}_2 \times \beta_2 - \cdots - \hat{\beta}_n \times \beta_n$$

$$Q = \sum_{i=1}^{n} e^2 = \sum_{i=1}^{n} (y_i - \hat{y}_i)^2 = \sum_{i=1}^{n} (y_i - \hat{\beta}_0 - \hat{\beta}_1 \times \beta_1 - \hat{\beta}_2 \times \beta_2 - \cdots - \hat{\beta}_n \times \beta_n)^2$$

Solving the partial guide of $\hat{\beta}_0,\ \hat{\beta}_1,\ \hat{\beta}_2, \cdots ,\ \hat{\beta}_n$,

$$\begin{cases} \frac{\partial q}{\partial \hat{\beta}_0} = 2 \times \sum_{i=1}^{n} (y_i - \hat{\beta}_0 - \hat{\beta}_1 \times \beta_1 - \hat{\beta}_2 \times \beta_2 - \cdots - \hat{\beta}_n \times \beta_n) \times (-1) = 0 \\ \frac{\partial q}{\partial \hat{\beta}_1} = 2 \times \sum_{i=1}^{n} (y_i - \hat{\beta}_0 - \hat{\beta}_1 \times \beta_1 - \hat{\beta}_2 \times \beta_2 - \cdots - \hat{\beta}_n \times \beta_n) \times (-\beta_1) = 0 \\ \vdots \\ \frac{\partial q}{\partial \hat{\beta}_n} = 2 \times \sum_{i=1}^{n} (y_i - \hat{\beta}_0 - \hat{\beta}_1 \times \beta_1 - \hat{\beta}_2 \times \beta_2 - \cdots - \hat{\beta}_n \times \beta_n) \times (-\beta_n) = 0 \end{cases}$$

Thus, $\hat{\beta}_0,\ \hat{\beta}_1,\ \hat{\beta}_2, \cdots ,\ \hat{\beta}_n$ are solved, so

$$x_{ij} = \hat{\beta}_j + \frac{\beta_0}{n \times \beta_i} i = 1,2,3, \cdots m;\ j = 1,2,3, \cdots n$$

Constantly determines the predicted value and adjusts the parameter, until the difference between the predicted value and the expected value ends within the allowable interval.

*4.1. Clustering and Parameter Adjustment*

The traditional Elman network uses the gradient descent method to train network parameters, but the gradient descent method has the disadvantages of slow convergence and easy to fall into local minimum values. Therefore, we use clustering and parameter adjustment methods to train network data, and achieve the goal of rapid optimization. And the trained model, that is, the improved routing decision is stored in the SDN controller. When there are new requests, the path can be quickly predicted to achieve SDWSN network load balancing.

Firstly, the training samples are fuzzy clustered, and they are subordinate to each category. These memberships can classify the training samples and construct a neural network classifier. For new unknown samples, it is not necessary to repeatedly calculate the clustering process. The category of the sample can be identified directly by the classifier. The process of clustering is as follows:

1. Enter link matrices {$TM_i$, i = 1, 2, ... , N}, and select initial cluster centers { $TM_i$, $TM_2$, ... , $TM_{Nc}$}, it may not be equal to the number of cluster centers required. The initial value can be arbitrarily selected from the sample.

2. The N traffic matrix is assigned to the type $S_j$, for the non-cluster center traffic matrix TM in the sample.

$$D_i = min\{\|TM - TM_i\|, i = 1,2, \cdots , N_c\}$$

If $D_j = \|TM - TM_j\|$ is the minimum value, then $TM \in S_j$.

3. If $S_j < \theta_N$, cancel the sample subset, then subtract 1 from $N_c$.

4. Correcting Cluster Centers

$$Z_j = \frac{1}{N_j} \sum_{TM \in S_j} TM j = 1,2, \cdots , N_c$$

5. Calculate the average distance between the traffic matrix and cluster centers in each clustering domain $S_j$

$$\overline{D_j} = \frac{1}{N_j} \sum_{TM \in S_j} \|TM - Z_j\| j = 1,2, \cdots , N_c$$

6.  Calculating the average of the distances between clusters of all link matrices and the corresponding clustering domain.

$$\overline{D} = \frac{1}{N} \sum_{j=1}^{N} TM_j \overline{D_j}$$

7. Splitting, merging and iterative operations
● If the number of iterations has reached I, that is, the last iteration, go to 11.
● If $N_c \leq \frac{K}{2}$, that is, the number of cluster centers is less than or equal to half of the initial value, go to 8 and split the existing clustering domain.
● If the number of iterative operations is an even number, or $N_c \geq 2k$, do not split, go to 11; otherwise, go to 8.

8. Calculate the Standard Deviation Vector of Sample Distance in Each Cluster.

$$S_j = (S_{1j}, S_{2j}, \cdots, S_{nj})^T$$

The components of the vector are $S_{ij} = \sqrt{\frac{1}{N_j} \sum_{k=1}^{N_j} (TM_{ik} - Z_{ij})^2}$ i = 1, 2, ... , n; j = 1, 2, ... , $N_c$.

9. Find the maximum component of each standard deviation vector $\{\delta_{j\max}, j == 1, 2, \cdots, N_c\}$,

10. In any maximum component set $\{S_{j\max}, j == 1, 2, \cdots, N_c\}$, if $S_{j\max} > \theta_S$, and satisfies one of the following two conditions:
● $\overline{D_j} > \overline{D}$, and $N_j > 2(\theta_N + 1)$
● $N_c \leq \frac{K}{2}$ Then $Z_j$ is split into two new cluster centers, and $N_c$ plus 1.

11. Calculate the distance of all cluster centers

$$D_{ij} = \|Z_i - Z_j\| i = 1, 2, \cdots N_c - 1; j = i + 1, \cdots, N_c$$

12.  Compare the values of $D_{ij}$ and $\theta_c$, The value of $D_{ij} < \theta_c$ is arranged incrementally by the minimum distance.

13. Merge two cluster centers $Z_{ik}$ and $Z_{jk}$ with distance $D_{ikjk}$ to get the new center.

$$Z_K^* = \frac{1}{N_{ik} + N_{jk}} \left[ N_{ik} Z_{ik} + N_{jk} Z_{jk} \right] k = 1, 2, \cdots, L$$

14. If the last iterative operation (the L-th time), the algorithm ends.

*4.2. Adjustment of Link Matrices*

Through multiple Elman networks, the input of the link traffic $Y$, the output of the traffic matrix $\hat{X}$ does not necessarily satisfy the constraint of $Y = A \times \hat{X}$. Therefore, some adjustments must be made to $\hat{X}$ to obtain $\hat{X}_0$.

**Definition 5.** *K-L distance: Let the random variable x be in the two probability distributions $P_{(x)}$ and $q_{(x)}$. The K-L distance is defined as:*

$$I_{(Q\|P)} = \sum_{i=1}^{n} P_{(x)} \times \log \frac{P_{(x)}}{q_{(x)}}$$

*where i represents the i − th value of x and p, q is the probability density function.*

**Definition 6.** *I-Projection: Given a joint probability distribution set $Q_{(x)}$ and $P_{(x)}$, if $Q_{(x)} \in Q$, and satisfy*

$$I_{(Q\|P)} = I_{(Q\|P)}$$

$Q_{(x)}$ is I-Projection of $P_{(x)}$. For a joint probability distribution set, if the constraint condition $p_{(Y)}$ is satisfied, I-Projection of $p_{(Y)}$ on $R_{(Y)}$ as follows:

$$Q_{(x)} = \begin{cases} 0 & P_{(Y)} = 0 \\ P_{(x)} \times \frac{R_{(Y)}}{P_{(Y)}} & P_{(Y)} \neq 0 \end{cases}$$

By iterating the constraints, the probability of all elements is estimated by minimizing the K−L distance. Through iterative calculations, adjustments are made while the constraints are met, so that the final result satisfies the maximum likelihood estimate.

*4.3. Parameter Adjustment*

The link traffic changes of the input layer of multiple Elman networks are random, and the parameters of multiple different Elman networks need to be adjusted. It is assumed that the validity periods $T$ of a plurality of different Elmans are the same. When $t < T$, the flow matrix components for a short period of time when $t_1 - t_2 = \Delta t \rightarrow T$ is selected are input for re-clustering, prediction, and comparison with the current clustering result. The grouping of changes in classification indicates that the current multi-Elman network is not applicable and needs to be adjusted, otherwise no adjustments.

## 5. Performance Evaluation

In this paper, the SDSNLB algorithm was simulated and tested by Matlab. In the simulation test, the wireless sensor network model consists of one center controller and three controllers, each controller is connected to control 10 sensors. In the subsequent tests, the number of sensor nodes was increased from 10 to 26, and the messages in the topology were fed back. The controller was operating normally and was able to process and process each message. In practical applications, the node layout is more dispersed, the number of hops from the node to the controller is different, and the message that the node interacts with the controller is concentrated at the same time is extremely unlikely, so the controller can efficiently process the incoming message.

In order to better demonstrate the effectiveness of the improved algorithm, the SDSNLB algorithm, the LEACH and improved LEACH protocol algorithm were compared and simulated. The LEACH protocol is a classic clustering-based routing protocol proposed in the WSN. Despite the shortcomings, studying the basic clustering protocol LEACH can help to better understand other clustering protocols. In addition, the hierarchical organization of the LEACH protocol is suitable for the dynamic distribution of resources. The cluster head node in the cluster can be regarded as a controller, and the cluster member node can be regarded as a node user, thereby better performing simulation. The improved LEACH algorithm is designed to extend network lifetime and improve efficient data transmission through balanced elections. The main analysis and comparison of the three algorithms on the network node link bandwidth utilization, network throughput, and network jitter.

Figure 4 shows the experimental results of the data flow under three different algorithms. As can be seen from the figure, when the conventional LEACH protocol algorithm is used, in a period of 0 to 10 s, the data flow in the network is relatively simple. Therefore, the effective throughput of the data stream is close to the uplink rate. And after 10 s, as the data traffic increases, the effective throughput of the data stream is significantly reduced. For the improved LEACH algorithm, the effective throughput of the data stream is close to the uplink rate from 0 to 20 s, and after 20 s, the effective throughput decreases. When using the SDSNLB algorithm, the effective throughput of the data stream approaches the uplink rate in 0 to 100 s.

In Figure 5, it can be seen that when there are multiple data streams in the network, the load balancing algorithm proposed in this paper can reasonably schedule the data flow according to the load of the actual gateway egress node, and the final traffic does not exceed the uplink rate of the link. The average bandwidth utilization is maintained at around 1.

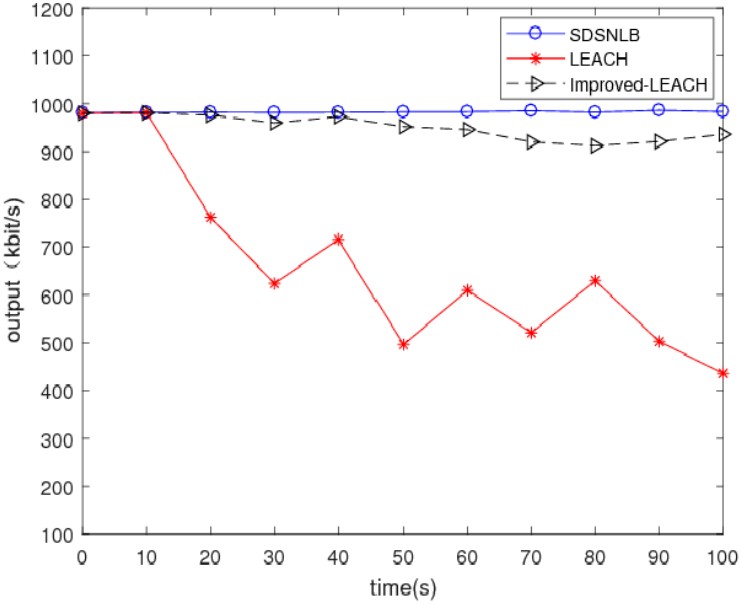

**Figure 4.** Data flow effective throughput.

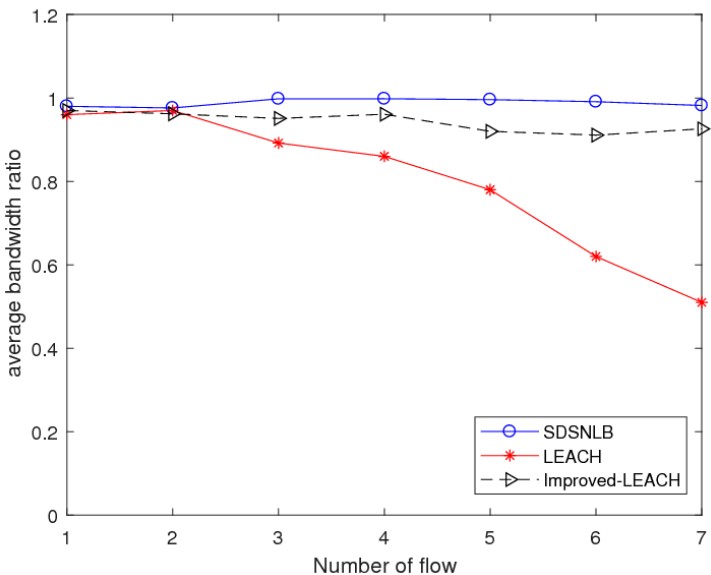

**Figure 5.** Average bandwidth utilization varies with the number of streams.

Figure 6 shows the performance of the link load jitter. When running LEACH protocol, the link load jitter is the highest. Compared with LEACH protocol, the standard deviation of link load in improved LEACH is smaller, but the link load jitter is still obvious. When the load balancing algorithm SDSNLB is running, the load standard deviation of all links in the network is minimum, the traffic load is more evenly distributed in the network link. Due to the load balancing algorithm proposed in this paper, it is possible to perform reasonable scheduling in real time according to the current distribution of traffic load in the network, so as to avoid premature traffic overload of individual transmission links in the network, thereby improving network performance.

From the above experimental results and analysis, the load balancing mechanism proposed in this paper can be verified. Comprehensive calculations were performed by an Elman neural network using software-defined features. The clustered feature data is transmitted to the controller node for fast load balancing of the sensor link, which can improve the average bandwidth utilization of the network and reduce the link load jitter of the network, thereby improving the overall performance of the network.

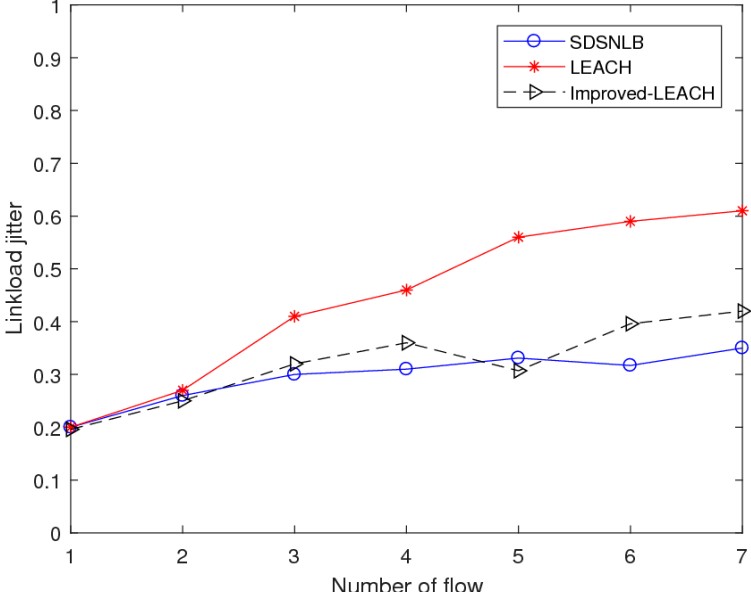

**Figure 6.** Link load jitter varies with the number of data streams.

## 6. Conclusions

The existence of a diversity set of paths for sensor nodes in SDWSN raises many technical issues related to data routing, where some routes become preferable for the nodes and lead to an imbalance. In order to improve the route optimization of the communication link and realize the load balancing of the network in the smart city sensor network, based on the characteristics of SDN centralized control and flexible traffic scheduling, the Elman neural network is used for the optimization calculation and the improved load balancing forwarding path is obtained. The simulation results show that the load balancing algorithm proposed in this paper can improve the average bandwidth utilization and reduce the link load jitter of the network, thus improving the performance of the entire network. It can also provide programmable data flow control method for software-defined sensor networks. The research in this paper has practical guidance and application significance for the smart city wireless infrastructure node link load balancing optimization problem.

**Author Contributions:** For this articles, individual contributions are as follows, conceptualization, X.H. and X.C.; methodology, X.C.; software, Q.M.; resources, Y.M.; writing—original draft preparation, X.C.; writing—review and editing, X.H., Y.M. and X.C.; funding acquisition, X.H. and X.C.

**Funding:** This research was funded by "Ministry of Education China Mobile Research Fund Project, grant number MCM20160304" and "National College Student Innovation Training Project, grant number 201810433023".

**Conflicts of Interest:** The authors declare no conflict of interest.

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
