# Peer review of "A Load Balancing Routing Mechanism Based on SDWSN in Smart City"

_electronics, doi:10.3390/electronics8030273_

Round 1

Reviewer 1 Report

The authors of this paper propose a new load balancing mechanism based on Software Defined Wireless Sensor Network (SDWSN) utilizing the advantages of centralized control of software defined networks and flexible traffic scheduling.

They use the OpenFlow protocol to monitor the running status and link load information of the network in real time.
Then the optimal load balanced routing is obtained by Elman neural network.
The authors compared their method with Low Energy Adaptive Clustering Hierarchy (LEACH) protocol and demonstrated better performance in balancing node traffic and imporving throughput.

The first paragraph of the 2.Related work section is too long and some parts need correction: Page 3, line 115-116: the ending of this sentence is confusing. Page 3, line 121-123: this sentence is confusing. Page 3, line 129: delete proposes. Page 3, line 133-134: this sentence is confusing. Page 3, line 139: this sentence is not needed. Additionally the authors might consider splitting this paragraph (lines 110-141) into smaller paragraphs. Page 4, Figure 2: the label of the center controller should change in English. Page 5, Table 1: spell correction of capacity at the meaning of the second parameter. Page 5, Table 1: capitalize the first word at the meaning of the tenth parameter. Page 6, Table 1: split sentence before 'if it ...' at the meaning of the second parameter (as shown in page 6).

Author Response

Dear reviewer:

       I am very grateful for your comments on the manuscript. Based on your suggestion, we modified the relevant sections of the manuscript and answered the questions. Please review the uploaded documents.

Reviewer 2 Report

The paper presents a load balancing routing mechanism for wireless sensor networks, based on SDWSN, and compares it to LEACH.

The paper needs extensive revision by a native English speaker as in its current form has parts that are not comprehensible (for example lines 63-65, lines 133-134, .

The authors should also present more clearly the relationship between the optimization solution step (in subsection 4.1) and the Elman neural networks. The overall scheme needs a better verbal explanation and possibly an example to make its workings clear.

The introduction contains elements of related work (e.g. lines lines41-53, and lines 63-76), while the related works section contains introductory text (e.g. lines 110-123). I suggest that the authors reorganize the text to have the problem description and any other relevant introductory text in section 1 and the related work in section 2.

All citations need careful checking. You cite some works by the first name of the first author, some with both by the first and last name; while it is customary to use the last name of all authors (and abbreviate with “et al”, if more than three). Most probably, the authors are mixing the first and last name of the authors.

Some other citation read “Literature” or “This review article” or something similar. Please use consistent citation style. In one citation, you say “Ariman and his team”. I am not sure that the other authors are the team of the first author (the Institution’s staff page list only the last author as an ass. prof.). Please use the appropriate citation style.

Also, please check carefully the references. A lot of references do not list all authors. I understand that this maybe a references style issue, but I believe that all authors should appear in the references and “et al” should be used when citing. Even if this style is ok, many references have only “et” instead of “et al”. References [5] and [15] are the same. Reference [7] is missing “… Raspberry Pi of …” in the title. Also “tested” should be “testbed”. Please take care with the references. Such sloppy presentation does not leave a good impression for the overall text.

Some of the related work presented concerns optical networks and I am not sure how relevant it is to the problem at hand. Please either document the relevance or consider excluding these works if irrelevant.

The paper talks about optimal and near optimal solutions to the problem. Still there is no proof of optimality. As I understand it the neural network selects a solution it calculates as best, but this is not what optimal means. Please consider rephrasing.

Please consider simplifying your notation. The average utilization of links is not dependent on link e and should not have € as a subscript.

Equation in line 215 is using k>j in a sum but no K in the summed terms. The same equation make no sense unless you define the ordering of the nodes (i.e. how you number the nodes). I understand that the equation means to say that what traffic goes in nodes should also go out of nodes (except for s and t), but this is not the correct way to say it. Please revise.

In line 223 you mention the receiving layer, but no such layer is described for the Elman neural net. Do you mean the inheritance (context / state) layer?

As far as I understand, lines 227-247 describe how the Elman neural network works. This does not seem to be a contribution of the paper but rather an explanation of the workings of the neural net. If my understanding is correct, you may consider removing this part of the paper, or briefly summarize it.

The equation in line 243 use the partial derivative symbol (like the Greek letter theta) which is also use for the utilization of links. This can create confusion. Please consider using another symbol/letter for the utilization of links.

Figures 3 and 4 have the same variable in the Y-axis but different scales. Please provide the units and explain the scales.

You compare your scheme (SDSNLB, but at the end of the introduction you name it SDWSNLB) to LEACH, which is clearly not LB. So this comparison does not really make sense and does not provide an indication of how better your proposed algorithm is with respect to other LB algorithms.

Overall, I find that the paper needs to be significantly revised, before reconsidering it for publication.

Author Response

(The authors gave the same response as above.)

Reviewer 3 Report

Summary: This paper proposes a load balancing mechanism of the multipath routings of the software defined wireless sensor networks. The basic idea is to solve the fairness optimization problem by using the Elman neural network. 

The contributions of the paper are not clear. Why do you use Elman neural network? The authors must clarify the reasons to apply the neural network for the optimization problem. What about the complexity of the proposed optimization problem? How is the convergence of the proposed scheme? 

In general, the paper is not well written. There are several grammatical errors and typos in this paper. The paper really needs the proofreading. Please maintain the consistent writing styles such as acronyms definitions. Furthermore, please use the acronyms after you define it instead of repeatedly defining it. 

After the LEACH protocol is proposed in 2000, there are many improved versions of the protocol. The performance evaluation of the proposed scheme is very weak. As the authors discuss in the related works, recent literature proposes better solutions on a similar problem.

Please describe better the optimization problem on page 6. Is it a min-max optimization problem? Please check the objective function. 

Since the authors apply the existing Elman neural network for clustered network designs, I suggest to include the background section regarding on Elman neural network and OpenFlow.  

Please describe better Fig 1. What does the interface name mean?

Some technical terms are not properly defined, e.g., theory system. 

Please remove Chinese letters in Fig. 2. 

Please add some references to motivating the problems in Section 1, “Introduction”. 

[1] R. Du, P. Santi, M. Xiao, A. V. Vasilakos and C. Fischione, "The sensable city: A survey on the deployment and management for smart city monitoring," in IEEE Communications Surveys & Tutorials.

[2] P. Park, S. Coleri Ergen, C. Fischione, C. Lu and K. H. Johansson, "Wireless Network Design for Control Systems: A Survey," in IEEE Communications Surveys & Tutorials, vol. 20, no. 2, pp. 978-1013, Secondquarter 2018.

[3] T. Qiu, N. Chen, K. Li, M. Atiquzzaman and W. Zhao, "How Can Heterogeneous Internet of Things Build Our Future: A Survey," in IEEE Communications Surveys & Tutorials, vol. 20, no. 3, pp. 2011-2027, thirdquarter 2018.

Author Response

(The authors gave the same response as above.)

Reviewer 4 Report

This paper proposes a load balancing scheme in SDWSN environment. However, the overall explanation is difficult and ambiguous. In particular, the authors should resolve the ambiguity below. 1. In Figure 2, Chinese characters need to be modified in English 2. In Table 1, yi is defined twice (ouput value and expected value) 3. Smart City is emphasized in the title, but there is no definition of Smart city and its relation with this proposed protocol is not clear. That is, Motivation, Introduction, Problem definition are ambiguous 4. There is insufficient analysis of previous research related to load balancing in related work section. Clear analysis and comparison with existing studies is needed. 5. Similar concepts are used in combination. (Load, traffic link utilization, connection weight, traffic matrix, and capacity). The authors shall explain the clear definition and differences of these terms, and how their numerical values are measured. 6. In addition to above comment #5, the authors shall provide a clear definition and calculation method for all the parameters shown in Table 1. 7. In section 5, detailed description of various experimetal environment such as specification of node configuration, topology structure, kind of data is needed. 8. The authors should also provide experimental result on the verification of the effect of load balancing, not the mere measurement of bandwidth and throughput. 9. Overall grammar correction through native speaker is required.

Author Response

(The authors gave the same response as above.)

Round 2

Reviewer 2 Report

The authors addressed this reviewer's comments, however not satisfactorily in all cases. In some cases the authors did not address the relevant issue but only provided explanations. Be reminded that the reccomendation was for major revision (emphasis on major).

My remaining concerns are:

a) It is still not quite clear how the heuristic and the Elman networks work together. I understand that the heuristic is used to do some calculations for input that pertains to real cases in the network, and then these are used to train the Elman network. This is done because the heuristic takes a long time to run and a trained Elman is able to respond quickly. However, is not clear when the Elman network is trained. Is it done once ? Is it done periodically? Is it doen continuously? Also, it is not clear which real cases are used to train the Elman network. You obviously cannot use all the cases/states appearing in the network (since the heurisrtioc takes a long time). How do you choose the real cases to make the calculations (using the heuristic) and feed the training of the Elman network? How does this chossing affect the performance of the Elman (i.e. how do you make sure that you train the Elman for all tytpes of situations that may arise in a real setting)?

When addressing this comment please consider that the reader must have enough information to be able to replicate your research. The description will be clear and sufficient only if a reader has enough understanding and information to do that.

b) You still talk about optimal and near-optimal solutions. Still you do not prove any kind of optimality. Unless you prove your heuristic to be optimal or near-optimal you should refrain from using that term.

c) You explain why you compare SDSNLB to LEACH, but you do not address the undelying concern. This comparison says nothing about how good SNSNLB is with respect to other schemes. You should present such a comparison to make the case that SDSNLB is useful. If you cannot do it directly, consider making it indirectly, by presenting how other schemes compare to LEACH (e.g., if another scheme is approx. 10% better than LEACH and yours is 20% better than LEACH, then yours is better that the other scheme).

Finally, the text still needs to be checked for the usage of the English language.

Author Response

Dear Reviewer,

I am very grateful to your comments for the manuscript. According with your advice, I have revised the manuscript again. Some of your questions were answered in Word file.

Reviewer 3 Report

Please rewrite the introduction of the paper. The authors may move the related works of [1, 2, 3] after the motivation of the problem in the introduction. There are some typos and grammatical errors of the paper.

Author Response

(The authors gave the same response as above.)

Reviewer 4 Report

The authors have improved the quality of the submitted paper by reflecting the opinions of the reviewer.

However, the configuration of the parameters (Response for Comment 6) and the verification environment (Response for Comment 7), which are the core of the proposed method, are not clearly shown.

In other words, the proposed method/scheme of the paper should be sufficient for the reader to reproduce and verify the performance, but the current explanations of manuscript do not help the reader.

Author Response

(The authors gave the same response as above.)

Round 3

Reviewer 2 Report

I am not satisfied with the authors response to my commnets. It seems that the authors did not take my suggestion for major revision ninto account and only provided simple additions changes to the text.

Concerning point 1 of my previous review, I feel that the authors did not answer the main quetsion, of how the Elman networks and the heuristic are used. The authors did not explain when the Elman networks are trained, and when this trining process is repeated. Please take a careful look at my previous comments (point 1) and try to explain in simple terms how your scheme works, how the Elman networks are used, when they are trained, how the heuristic is used, when it is used etc. The method you proposed should be understandable by any reader.

Concerning point 3, the authors only included Improved-LEACH into the figures. However, this shows that the concern for comparing SDNLB onwith LEACH is valid. The improvement over LEACH means nothing. Looking at the improvements over Improved-LEACH, it seems that SDSNLB is marginally better. The authors do not discuss this. They only pinpont the improvement of=ver LEACH. Still the marginal improvement over Improved-LEACH show that SDSNLB should be compared with other schemes to demonstrate that it is better than the state-of-the-art. I can understand that direct comparison might ne dfficult. Still the authors should at least show that other schemes do not have a better improvement over Improved-LEACH.

I am afraid tha the authors have not considerably improved the text over the previous version. I am still recommending major revision, but the authors should consider thsi as an opportunity and focus seriously into addressing the comments.

Author Response

(The authors gave the same response as above.)

Reviewer 4 Report

The authors have improved the quality of the paper by faithfully reflecting the opinions of the reviewer. So the only comment is as follows.

Since the resolution in Figures 4, 5, and 6 is somewhat low, it is recommended to revise it to the vector type picture.

Author Response

(The authors gave the same response as above.)
